

# Kinematic effects of the circle with and without rider in walking horses

Agneta Egenvall[1], Hanna Engström[2] and Anna Byström[3]

[1] Department of Clinical Sciences, Swedish University of Agricultural Sciences, Uppsala, Sweden
[2] Ekeskogs Riding Academy, Ekeskogs Riding Academy, Klintehamn, Sweden
[3] Department of Anatomy, Physiology and Biochemistry, Swedish University of Agricultural Sciences, Uppsala, Sweden

## ABSTRACT

**Background:** Biomechanical studies of walk, especially walk on the circle, are scarce, while circles or curved tracks are frequently used during equestrian activities. To study horse–rider–circle interactions on the circle, the first steps would be to investigate how the unridden, freely walking horse is influenced by circular movement, and then add a rider. The aim was to study horse vertical trunk movements, and sagittal cannon angles (protraction–retraction) during walk in straight-line and on the circle without rider, and on the circle with a rider using minimal influence.

**Methods:** Ten horses were ridden by five riders, summing to 14 trials. Each trial included straight walk unridden (on concrete), and walk on 10 m diameter circles (left and right on soft surface) first lunged (unridden) and then ridden with minimal rider influence. Inertial measurement units (100 Hz) were positioned on the withers, third sacral vertebra (S3) and laterally on metacarpal and metatarsal bones (using self-adhesive bandage). Selected data were split in steps (withers and S3 vertical translations) or strides (cannon protraction–retraction) at maximum hind limb protraction, and range of motion (ROM), minima and maxima, and their timing, were extracted. Data were analyzed using mixed models with inner/outer/straight nested within unridden/ridden as fixed effect, and controlling for stride duration. Differences between: inner vs outer steps/limbs; the same step/limb unridden vs ridden; and the same step/limb straight vs inner/outer unridden; were examined for statistical significance at $p < 0.05$.

**Results:** Inner limbs had smaller cannon ROM than outer limbs, for example, forelimbs when ridden (inner vs outer 62° vs 63°) and hind limbs when unridden (53° vs 56°). Forelimb cannon ROM was the largest for straight (65°). Hind limb ROM for straight walk (55°) was in-between inner (52–53°) and outer hind limbs (56–57°). Vertical ROM of S3 was larger during the inner (unridden/ridden 0.050/0.052 m) vs the outer step (unridden/ridden 0.049/0.051 m). Inner (0.050 m) and outer steps (0.049 m) unridden had smaller S3 ROM compared to straight steps (unridden, 0.054 m). Compared to when unridden, withers ROM was smaller when ridden: inner hind steps unridden/ridden 0.020 vs 0.015 m and outer hind steps 0.020 vs 0.013 m. When ridden, withers ROM was larger during the inner hind step vs the outer.

**Conclusion:** The outer hind limb had greater cannon pro-retraction ROM, compared to the inner limb. Larger croup ROM during the inner step appears to be coupled to increased retraction of the outer hind limb. Knowledge of magnitudes and

Corresponding author
Agneta Egenvall,
agneta.egenvall@slu.se

timing of the horse's movements on the circle in unridden and ridden walk may stimulate riders to educate eye and feel in analyzing the execution of circles, and stimulate further studies of the walk, for example, on interactions with rider influence, natural horse asymmetries, or lameness.

## INTRODUCTION

The straight walk is a symmetrical four-beat gait with alternating bipedal and tripedal support phases, that is, either two limbs or three limbs are in contact with the ground (*Gan et al., 2016*). This results in out of phase movement of the fore- and hindquarters. For example, when the two forelimbs are at early and late stance, respectively, the withers will be at its lowest and the head and croup at their highest; these events coincide with hind limb midstance. The saddle has two periods of pitch during each stride: the front part of the saddle is lowered from the dual hind limb support to the dual forelimb support. In roll and yaw the saddle has only one period of rotation per stride, tilting and turning (the rear part) away from the hind limb in stance. Yaw motion of the saddle, as well as pitch of the horse's trunk, are larger in walk compared to in trot (*Byström et al., 2010*). When schooling the walk, the movement characteristics of the walk makes it challenging for the rider to ensure that the horse maintains a regular four-beat gait (*Hess et al., 2012*).

*Chateau, Degueurce & Denoix (2005)* studied the effect of a sharp turn in walk on asphalt in four horses. They reported that the inside forelimb adducted progressively throughout the stance phase, advancing the body over the limb in the direction of motion. *Hobbs, Licka & Polman (2011)* found that stride length was smaller for inner compared to outer limbs, studying six horses lunged in walk on 10 m diameter circles on soft surface. Neither of these kinematic studies included measurements on straight line, however, that is, no direct comparisons were made between straight-line walk and circle. *Wakeling et al. (2006)* studied 26 horses walking in hand in a figure-eight on 5 m circles. Electromyographic measurements of the left and right of *m. longissimus dorsi* showed that muscle activity was four to five times greater on the inside compared to at the straight, and that the outside muscle activity decreased compared to straight.

Riders will strive to achieve straightness (*Hess et al., 2012*), which means that the shape of the horse's spine should conform with the current track, whether curved or straight, and that each (ipsilateral) fore- and hind limb pair should follow the same track at all times. The goal of straightness training is to remedy the natural crookedness of the horse, and achieve symmetrical muscle development and equal suppleness to the left and to the right, to benefit both locomotor health and performance. The rider is advised to begin this work already with the young horse, start by riding on lightly curved and continuing to more curved tracks while continuously striving to achieve equal bending, rein contact and hind limb engagement on both hands (*Hess et al., 2012*). Thus, riding with a

bend or on a circle is common (*Eisersiö et al., 2015*), and it is of interest to study the interplay between horse and rider in this situation. At straight walk it has been shown that the rider can influence the movement pattern of the horse (*Weishaupt et al., 2006*; *Rhodin et al., 2018*), but for walk on the circle there are neither any studies comparing between ridden and unridden walk, nor any studies on horse-rider interactions. In order to understand the interaction between the horse, the rider and the circle, the first step would be to investigate how the unridden, freely walking horse is influenced by circular movement, and thereafter to add the rider to the equation.

The aim of the current study was to explore differences in vertical excursion of the forehand and hindquarters, and in limb pendulation, between inner and outer limbs, in horses walking on a circle unridden, and ridden with minimal rider influence, as well as unridden in straight line. The study focused on vertical translation of the withers and the third sacral vertebra (S3), and sagittal metacarpal and metatarsal bone angles, approximating limb protraction and retraction. The hypotheses were that variable values for straight walk would be in-between those for outer and inner steps, that angular ROM (protraction–retraction) of the cannon bones of the outer limbs would exceed those of the inner limbs, and that the addition of a rider would influence the forehand variables more than hind quarter variables.

## MATERIALS AND METHODS

### Horses and their equipment

The experiment took place at one stable, using 10 horses that had been stabled there for at least 2 months. Data on breed, gender, age, withers height, and duration with the present owner were assembled (Table 1). The horses were trained in classical dressage, varying from basic to advanced level. The horses were deemed sound (<1 of 5 on the AAEP scale, *AAEP Horse Show Committee, 1999*) by a veterinarian (AE) who watched the horses for at least one full session of ridden work in walk and trot, in proximity to the experiment. The horses were equipped with their custom saddle (the weight of the saddles varied from 7 to 9 kg, including stirrups, stirrup leathers and girth; one horse was ridden bare-back) and head gear, for details see Table 1. A cavesson was mounted to be used while lunging and one horse was also ridden (bitless) on the cavesson. According to Swedish law this study did not need specific ethical approval. Riders provided written consent.

### Riders and their equipment

Five right-handed riders participated; one of them was a trainer and the others her students. The riders were all female, aged between 18 and 57 years, with weights and heights ranging between 61–66 kg and 158–173 cm. The riders had been training regularly with the trainer for at least 4 years. All riders were accustomed to their mounts. Riders wore helmets and all carried a dressage whip (rarely used during the experiment), but spurs were not used. All riders except one rode two or more horses. Four horses were ridden by two riders (Table 1). There was no intention to achieve a complete cross-over of horses and riders.

**Table 1 Details on the 10 horses in the study, ridden by five riders.**

| Horse no. | Breed | Gender | Age (yrs) | Height at withers (cm) | Duration with owner (yrs) | Rider no. | Saddle | Bridle |
|---|---|---|---|---|---|---|---|---|
| 1 | Russian crossbred/PRE[a] | Gelding | 17 | 158 | 3 | 1,3 | Acad saddle[b] | Kimberwick, straight bar |
| 2 | Lusitano | Gelding | 11 | 150 | 2 | 1,3 | Acad saddle | Snaffle, straight bar, D-rings |
| 3 | Knabstrupper | Mare | 11 | 155 | 7 | 2 | Acad saddle | Baucher |
| 4 | PRE | Mare | 13 | 158 | 4 | 2 | Acad saddle | Baucher |
| 5 | Lusitano | Mare | 16 | 154 | 10 | 1,3 | Acad saddle | Curb bit |
| 6 | Newforest | Gelding | 15 | 138 | 4 | 4,5 | Bareback | Bitless, cavesson |
| 7 | PRE | Mare | 17 | 158 | 13 | 1 | Acad saddle | Kimberwick, straight bar |
| 8 | Lusitano | Mare | 7 | 155 | 1 | 5 | Acad saddle | Snaffle |
| 9 | PRE | Stallion | 11 | 153 | 10 | 1 | Acad saddle | Snaffle with eggbutt rings |
| 10 | PRE/Frederiksborg horse | Mare | 11 | 158 | 11 | 3 | Acad saddle | Kimberwick, straight bar |

**Notes:**
[a] PRE, pura raza Española.
[b] Academic saddle.

## Equipment

Inertial measurement units (IMU) (NGIMUs; x-io Technologies Limited, Bristol, UK, https://x-io.co.uk/ngimu/) were positioned on the withers, above the third sacral vertebra (S3), and laterally on the metacarpal and metatarsal (cannon) bones of both fore- and hind limbs, using self-adhesive bandages (Fig. 1). The sign conventions were that for midline sensors linear acceleration was positive upwards, and for cannon sensors pitch was positive clockwise when seen from the horse's left side, that is, equivalent to protraction. The size of the IMUs was $56 \times 39 \times 18$ mm, weight 46 g and resolution 16 bit. The sampling rate was set to 100 Hz. The IMUs were synchronized using proprietary software (https://x-io.co.uk/ngimu/). The experiments were video-recorded (Sony FDR-AX53) at 25 Hz; when recording trials on the circle the camera was placed in the middle of the circle, and head/tail view was used for straight line. In order to synchronize the video with the IMU measurements, the sensors were tapped before and after each trial, positioned together on a stool placed in front of the camera.

## Design

After starting the video camera, tapping and then placing the sensors, and tacking up the horse, the horses were first walked in hand on concrete in a straight alley, striving for a brisk walk. The horses were either led from in-front, or at a distance of 1.5 m from the horse's side, to minimize influence from the handler. The handler was either the rider, or researchers AE or HE. A 10 m diameter circle track was then marked on a soft sand-fibre surface in an indoor arena ($20 \times 30$ m). The horse was then lunged on the marked circle by the rider in free walk until the horse had completed at least one circle in steady walk per direction. After that the rider mounted and rode the horse in free walk on the circle. When the trial was finished the sensors were removed, tapped for

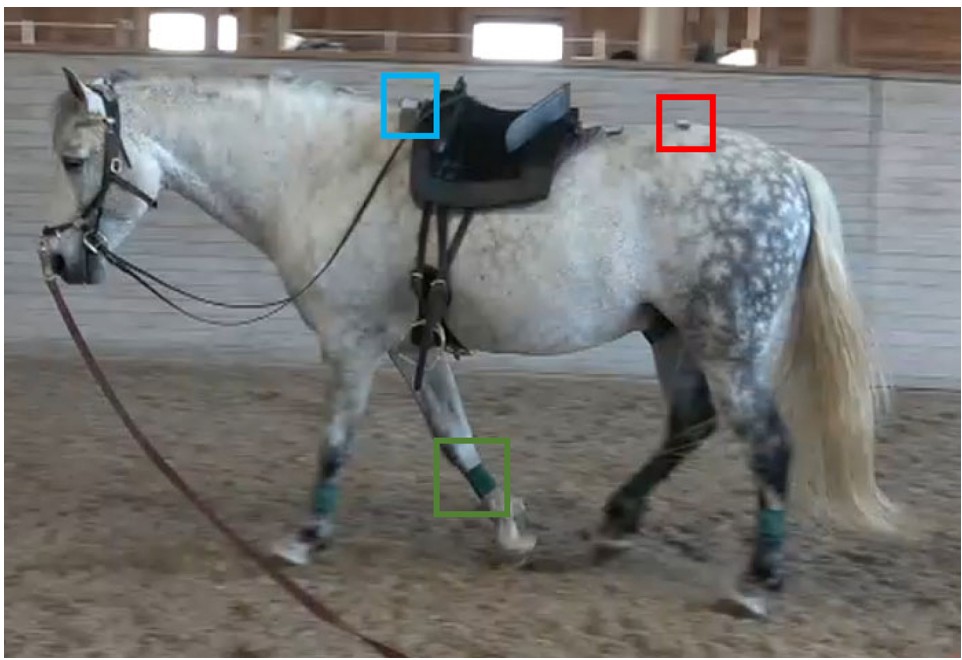

**Figure 1 A horse with sensors.** A study horse (horse 9) with sensors mounted to the withers (blue rectangle), the third sacral vertebra (red rectangle) and laterally on the four cannons using self-adhesive wrapping (green rectangle on right forelimb).

synchronization, and then the video camera stopped. The rider rode with loose reins while still guiding the horse to follow the circle. Left and right leads were alternated, such that the horses were either lunged unridden in left-right lead order and ridden in right-left lead order, or in the opposite. Left/right order was consistent for all horses within each measurement day but was reversed between days. Warming-up between unridden and ridden conditions was minimal: walking from the mounting block to the circle (<20 strides). One to five trials were performed per day, but each horse only performed one trial per day. Four horses performed the experiment twice (with different riders), with approx. 5 month between. Only one horse was in the arena during data collection.

## Data analysis

The videos were scrutinized by authors AE and AB. Data handling was done using Matlab (Matlab version 2019b; The MathWorks® Inc., Natick, MA, USA). Straight line data were selected from two of the longest and straightest moving walk sequences out of 2–4 recorded. Data from the circle were selected as an uninterrupted sequence, usually 0.5–1 circle, when the horse was moving as freely and harmonious as possible and following the circle, with minimal active influence from the lunge line or the rider. Data from selected sequences were split into steps (midline vertical excursions, one inner and one outer step per stride) or strides (cannon angles) based on the maximum pitch angle for each hind limb cannon bone, approximating ground contact times of this
limb. When stride-split data were used, strides from circles were split on the inner hind limb cannon pitch maximum, and strides from straight walk on left hind limb cannon pitch maximum. Stride/step curves for all used steps were plotted and visually inspected. Duration (in s) of each stride was calculated. Data were time-normalized to 100 points/indices.

ROM, minima and maxima, and timing of those extreme points relative to hind limb cannon maximum pitch (protraction), were calculated from step-normalised vertical position data for the withers and S3, and from stride-normalised pitch data for the cannons. These were deemed the most relevant and reliable kinematic variables extractable from the IMU data. Maximum and minimum cannon pitch angles will be referred to as maximum protraction and maximum retraction, respectively, in the results section. Step/stride minima and maxima were extracted automatically; because the vertical position data for S3 and the withers (especially) were more noisy, outlier removal was applied as follows, to only include data with a correct outtake of extreme points: for withers minimum position and its timing, data points were only included if minima were found between 60% and 95% of the step, and for withers maximum only if between 10% and 50%. For S3 minimum and its timing, data were included if found between 10% and 50% of the step, and for S3 maximum between 50% and 100%. These constraints were selected based on the expected timing of those extreme points in a normal walk. For withers this resulted in that 33–46% of the data points were removed and for S3 data 2–11% of the data points were removed. Timing values were adjusted by 50% (of the stride) as necessary in order for values to be directly comparable between inner/outer/straight step or limb values. For straight line data left and right steps/limbs were then pooled.

## Statistics

Descriptive and analytical statistics were made using SAS (SAS Institute Inc., Cary, NC, USA, the MIXED procedure was used for the analytical part). Stride duration was entered as a linear covariate in all models after verifying that no curved or modal pattern could be suspected when plotting stride duration against the outcome variables. The general random parameter structure consisted of rider, horse, the interaction between horse and rider, and exercise (e.g., unridden on left lead) within rider-horse combination. The option pdiff (in the MIXED procedure) was used for extracting least square means and selected pairwise comparisons. The following two-way comparisons were evaluated: (a) inner vs outer step/limb variable values, separately for unridden and ridden conditions; (b) the same step/limb variable (e.g., inner hind limb cannon maximum protraction) between unridden vs ridden; and (c) the same variable between straight vs inner or outer (e.g., inner hind maximum protraction vs maximum protraction on straight). Straight was not compared to ridden. For minimum and maximum vertical positions only (a) comparisons were made (since IMUs do not measure actual height). The $p$-value limit for statistical significance was set to <0.05. $P$-values were not adjusted for multiple comparisons.

The following two groups of models were created:

(1) In order to study the vertical excursion of the trunk, withers and S3 ROM, and values for and timing of minima and maxima, were analyzed with each observation/step labeled as an inner, outer or straight step. The fixed categorical effect evaluated was step (inner/outer/straight) nested within unridden/ridden.
For minimum and maximum value evaluation an extra random effect was added: stride serial number nested within the three-way interaction between horse, rider and exercise, enabling the comparison of inner and outer step minima/maxima within condition (unridden/ridden) on a stride by stride basis.

(2) In order to study cannon bone protraction–retraction (pitch), ROM and values for and timing of minima and maxima were analyzed using stride-normalised data. The fixed categorical effect evaluated was limb (inner fore, outer fore, inner hind, outer hind, straight fore and straight hind) nested within unridden/ridden.
For cannon maximum protraction and maximum retraction angles an extra random effect was added: the three-way interaction between horse, rider and horse side (left/right) nested within exercise, to account for slight differences in sensor alignment with the cannon bone on left and right limbs.

Additionally, a model with stride duration as outcome was made, with the same general random parameter structure. Fixed effects were circle/straight nested within unridden/ridden, and hind limb cannon ROM. Stride duration was plotted vs hind limb cannon ROM to confirm reasonable linearity vs the outcome.

To achieve normality of residuals the following steps were taken. If mean and medians were close, standard deviations (SDs) relatively small and skewness and kurtosis close to zero, variables were analyzed in untransformed formats. If this was not the case, variables were Box Cox transformed and the best practical transformation for each variable was adopted. Residual plotting was employed in a last step to ensure that only few standardized (Pearson) residuals had absolute values above three.

## RESULTS

### Descriptive results

In total there were 14 trials (a trial defined as one complete measurement session for one horse) from 10 horses ridden by 5 riders, each riding 1–5 horses. Each trial included unridden walk in hand on straight line and on the lunge on left and right circles, and then ridden walk on circles. In the overall dataset, for each horse/rider combination (trial) there were between 13 and 26 strides of straight walk (median 18 strides), 4–49 strides of unridden walk on the circle to the left (median 13 strides), 6–25 strides of unridden walk on the circle to the right (median 12 strides), 4–34 strides of ridden walk on the circle to the left (median 7.5 strides) and 4–33 strides of ridden walk on the circle to the right (median 11 strides). Descriptive data for stride duration are found in Table 2. Figure 2 shows time-normalised data curves from one stride for one horse (unridden and ridden) on the circle, illustrating the relative timing of minima and maxima for the six variables analyzed. Figs. 3 and 4 show examples of time-normalised stride by stride data for all exercises.

**Table 2 Overview of stride duration results.**

| Exercise | N | Percentile | | | | |
|---|---|---|---|---|---|---|
| | | Mean | SD | 5 | 50 | 95 |
| Unridden left circle | 14 | 1.28 | 0.1 | 1.17 | 1.25 | 1.5 |
| Unridden right circle | 14 | 1.29 | 0.1 | 1.17 | 1.27 | 1.51 |
| Ridden left circle | 14 | 1.28 | 0.12 | 1.08 | 1.28 | 1.56 |
| Ridden right circle | 14 | 1.26 | 0.11 | 1.05 | 1.26 | 1.44 |
| Straight | 14 | 1.2 | 0.05 | 1.11 | 1.19 | 1.29 |
| Location | Model | Est | SE | *P*-value | | |
| Withers | ROM[a] | −0.016 | 0.003 | <0.0001 | | |
| S3 | ROM | −0.065 | 0.003 | <0.0001 | | |
| Cannon sagittal | ROM | −14.459 | 0.893 | <0.0001 | | |
| Pendulation | Max protraction[b] | 3.389 | 1.111 | 0.002 | | |
| | Max retraction | −10.891 | 1.005 | <0.0001 | | |

Notes:
[a] ROM, range of motion.
[b] Max, maximum.
The upper part of the table demonstrates stride duration (s) over the 14 trials in the study (based on trial means).
The lower part shows estimates (Est), and standard errors (SE), when stride duration was included as a covariate in the vertical excursion and limb pendulation models (the other results for these models reported in Tables 4 and 5).
For example, ROM of the withers was estimated to decrease by −0.016 m for 1 s increase in stride duration.

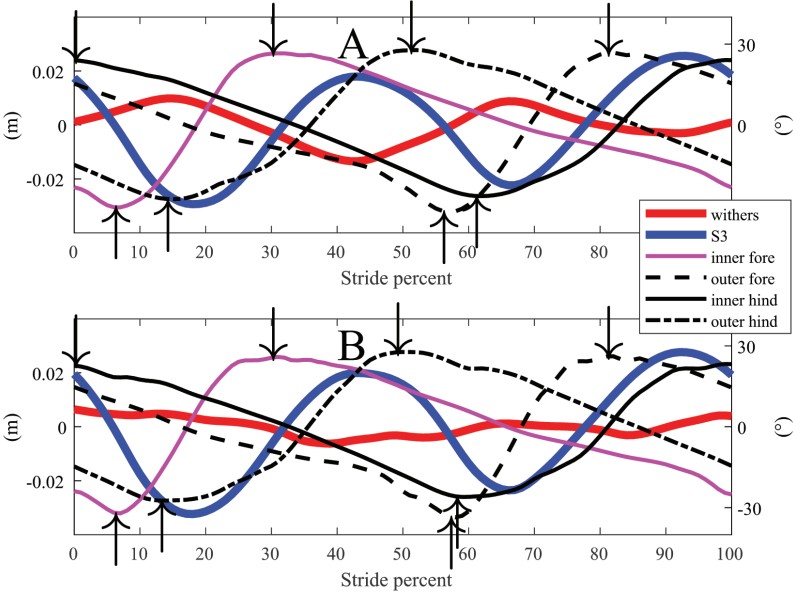

**Figure 2 Time-normalised data, representing one stride, from horse 1 unridden and ridden on the left circle.** Downward arrows represent maximum protraction and upward arrows maximal limb retraction. (A) Unridden. (B) Ridden.

## General modeling results

Data distributions, including number of steps or strides, for the variables analyzed are illustrated using boxplots in Figs. S1–S3. Variables were modeled in untransformed format, except timing variables for midline markers; timing of withers maximum position

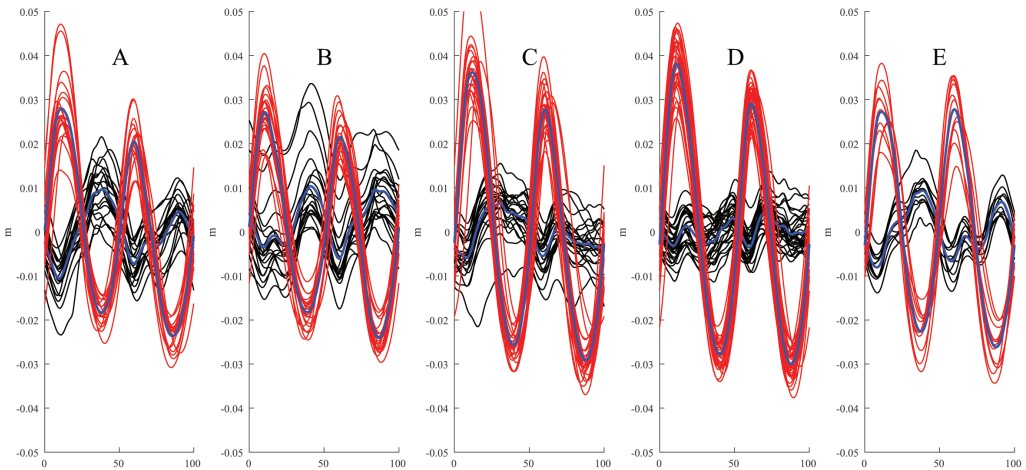

**Figure 3 Examples of time-normalised stride data for the withers (black traces/blue mean curve) and S3 vertical positions (red/blue) from one horse.** The data are split on the maximum protraction of the inner hind limb and the left hind limb when straight. (A) Unridden left lead. (B) Unridden right lead. (C) Ridden left lead. (D) Ridden right lead. (E) Straight.

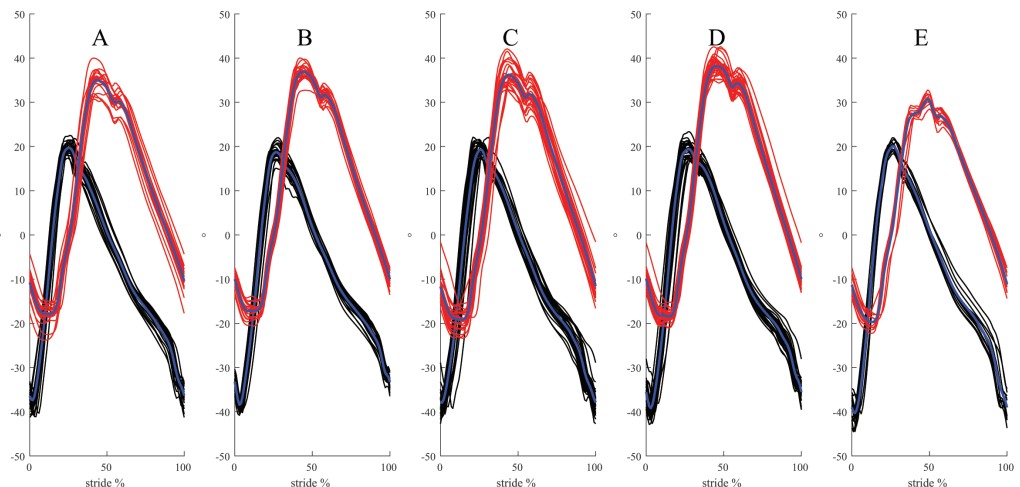

**Figure 4 Examples of time-normalised stride data for the outer fore limb (black traces/blue mean curve) and inner hind limb (red/blue) from one horse. For straight right forelimb and left hind limb are shown.** The data are split on the maximum protraction of the inner hind limb and the left hind limb when straight. (A) Unridden left lead. (B) Unridden right lead. (C) Ridden left lead. (D) Ridden right lead. (E) Straight.

was modeled squared, and other midline timing variables in square-root format. The contributions of the various random effects in each of the models can be found in the SAS-output in Table S1. In all midline and cannon models the fixed effects (stride duration and circle/straight within unridden/ridden) had type III $p$-values $p < 0.0001$, except for: in the timing of withers minimum model stride duration the $p$-value was 0.10; in the timing for withers maximum model circle/straight within unridden/ridden was associated with $p = 0.0005$; in the timing of maximum protraction model stride duration was associated with $p = 0.24$; and in the maximum protraction model stride duration

**Table 3 Summary of significant findings from the models.**

| | | Significantly larger or later in the comparison | | | | | |
|---|---|---|---|---|---|---|---|
| | | Inner | Outer | Unridden | Ridden | To straight | |
| | | Unridden/ridden | Unridden/ridden | Inner/outer | Inner/outer | Inner/straight (unridden) | Outer/straight |
| Forelimb | ROM[a] | | | | outer | straight | straight |
| | Maximum protraction | | | | | | |
| | Timing maximum protraction | | | | outer | | |
| | Maximum retraction | | | | | straight | straight |
| | Timing maximum retraction | | | outer | outer | straight | |
| Hind limb | ROM | | | outer | outer | straight | outer |
| | Maximum protraction | | | | outer | | outer |
| | Timing maximum protraction | | | inner | inner | | straight |
| | Maximum retraction | | | outer | outer | | straight |
| | Timing maximum retraction | | | inner | inner | | straight |
| Withers | ROM | unridden | unridden | | inner | | |
| | Maximum vertical position | NA | NA | | | NA | NA |
| | Timing maximal position | | | | outer | inner | outer |
| | Minimum vertical position | NA | NA | | inner | NA | NA |
| | Timing minimum position | | | inner | inner | inner | |
| S3[b] | ROM | | | inner | inner | straight | straight |
| | Maximum vertical position | NA | NA | outer | outer | NA | NA |
| | Timing maximum position | ridden | ridden | inner | inner | inner | |
| | Minimum vertical position | NA | NA | inner | inner | NA | NA |
| | Timing minimum position | ridden | | inner | inner | | |

**Notes:**
[a] ROM, range of motion.
[b] S3, third sacral vertebra.
Models are for withers and S3 vertical position, and forelimb and hind limb cannon (metacarpal/metatarsal bone) protraction-retraction; detailed model results are shown in Tables 4 and 5. For each pair-wise comparison (columns) and outcome variable (rows), the condition with the largest value, or latest stride percentage, is given ($p < 0.05$). The header states the conditions compared. Empty cells indicate no significant difference, and NA comparison not evaluated.

was associated with $p = 0.002$. Table 2 demonstrates stride duration coefficients for selected models (the rest in Table S1). A summary of the significant findings is found in Table 3. For each variable, for example, forelimb cannon protraction-retraction, a total of six comparisons were evaluated, except for withers and S3 maximum and minimum vertical positions, for which only two comparisons were made (inner vs outer step unridden and ridden).

Detailed results for withers and S3 vertical position variables can be found in Table 4 (bold indicates $p$-values $<0.01$ and italics $0.01 < p < 0.05$). For example, for both unridden ($p = 0.001$) and ridden conditions ($p = 0.03$), S3 ROM was larger during the inner step, compared to the outer step (Table 4). For interpretation of withers vertical position variables, it should be noted that the inner hind limb step includes first the maximum concurrent with outside forelimb midstance, and then the minimum at inside forelimb early stance (Fig. 2). For cannon angles (Table 5) there were no significant differences for the same limb (e.g. inner) between unridden and ridden. Inner limbs had smaller

**Table 4 Midline parameter models.**

| Variable | Step[a] | Ridden[b] | Est | SE | BT[c] | Comp[d] | | | Variable | Step | Ridden | Est | SE | BT | Comp | | |
|---|---|---|---|---|---|---|---|---|---|---|---|---|---|---|---|---|---|
| Withers | Inner | 0 | 0.020 | 0.001 | | **c** | – | – | S3 | Inner | 0 | 0.050 | 0.003 | | – | **c** | **c** |
| Vertical | Inner | 1 | 0.015 | 0.001 | | **c** | | **c** | Vertical | Inner | 1 | 0.052 | 0.003 | | – | | *c* |
| ROM[e] | Outer | 0 | 0.020 | 0.001 | | | **c** | – | ROM | Outer | 0 | 0.049 | 0.003 | | | – | **c** |
| | Outer | 1 | 0.013 | 0.001 | | | **c** | **c** | | Outer | 1 | 0.051 | 0.003 | | | – | *c* |
| (n = 1,180) | Straight | 0 | 0.019 | 0.001 | | | | – | (n = 1,895) | Straight | 0 | 0.054 | 0.002 | | | | **c** |
| Timing | Inner | 0 | 8.890 | 0.081 | 78.5 | – | **c** | **c** | Timing | Inner | 0 | 5.500 | 0.124 | 29.7 | *c* | **c** | – |
| Min[f] vertical | Inner | 1 | 8.826 | 0.086 | 77.4 | – | | **c** | Min vertical | Inner | 1 | 5.724 | 0.125 | 32.3 | *c* | | **c** |
| Withers | Outer | 0 | 8.701 | 0.082 | 75.2 | | – | **c** | S3 | Outer | 0 | 5.364 | 0.124 | 28.3 | | – | **c** |
| Position | Outer | 1 | 8.572 | 0.086 | 73.0 | | – | **c** | Position | Outer | 1 | 5.552 | 0.125 | 30.3 | | – | **c** |
| (n = 1,414) | Straight | 0 | 8.693 | 0.083 | 75.1 | | **c** | – | (n = 2,088) | Straight | 0 | 5.331 | 0.124 | 27.9 | | – | – |
| Timing | Inner | 0 | 5.042 | 0.109 | 24.9 | – | – | **c** | Timing | Inner | 0 | 7339 | 269.9 | 85.2 | **c** | **c** | **c** |
| Max[g] vertical | Inner | 1 | 5.017 | 0.118 | 24.7 | – | | *c* | Max vertical | Inner | 1 | 7869 | 270.4 | 88.2 | **c** | | **c** |
| Withers | Outer | 0 | 5.068 | 0.109 | 25.2 | – | – | **c** | S3 | Outer | 0 | 6895 | 269.7 | 82.5 | *c* | **c** | – |
| Position | Outer | 1 | 5.137 | 0.117 | 25.9 | – | | *c* | Position | Outer | 1 | 7362 | 270.4 | 85.3 | *c* | | **c** |
| (n = 1,467) | Straight | 0 | 4.691 | 0.111 | 21.5 | | **c** | **c** | (n = 1,900) | Straight | 0 | 6636 | 269.4 | 81.0 | | **c** | – |
| Min | Inner | 0 | −0.010 | 0.001 | | | – | | Min | Inner | 0 | −0.028 | 0.001 | | | *c* | |
| Vertical | Inner | 1 | −0.008 | 0.001 | | | | *c* | Vertical | Inner | 1 | −0.028 | 0.001 | | | | *c* |
| Withers | Outer | 0 | −0.009 | 0.001 | | | – | | S3 | Outer | 0 | −0.023 | 0.001 | | | *c* | |
| Position | Outer | 1 | −0.007 | 0.001 | | | | *c* | Position | Outer | 1 | −0.024 | 0.001 | | | | *c* |
| (n = 1,414) | Straight | 0 | −0.009 | 0.001 | | | | | (n = 2,088) | Straight | 0 | −0.028 | 0.001 | | | | |
| Max | Inner | 0 | 0.010 | 0.001 | | | – | | Max | Inner | 0 | 0.022 | 0.001 | | | **c** | |
| Vertical | Inner | 1 | 0.007 | 0.001 | | | | – | Vertical | Inner | 1 | 0.024 | 0.001 | | | | **c** |
| Withers | Outer | 0 | 0.010 | 0.001 | | | – | | S3 | Outer | 0 | 0.026 | 0.001 | | | **c** | |
| Position | Outer | 1 | 0.006 | 0.001 | | | | – | Position | Outer | 1 | 0.027 | 0.001 | | | | **c** |
| (n = 1,467) | Straight | 0 | 0.010 | 0.001 | | | | | (n = 1,900) | Straight | 0 | 0.026 | 0.001 | | | | |

**Notes:**
[a] Step relates to the hind limb step.
[b] Unridden (0), ridden (1).
[c] BT-back-transformed estimate.
[d] Comp (c)., comparison.
[e] ROM, range of motion.
[f] Min, minimum.
[g] Max, maximum.
Least square means estimates (Est) and standard errors (SE) for midline parameter data (in m). Timing results are reported as step percentages. Back-transformed estimates (BT) are reported in case a transformation was used. The number of observations for each model are shown in the table. The models include data from 10 horses, unridden, and ridden by 5 riders, in total 14 trials, however for the S3 data one horse did not have synchronized data (9 horses in the right-column analyses). Pairs of cells with '–' in the same column (headings Comp) indicate non-significant comparisons ($p \geq 0.05$), pairs of cells with c:s in italics; ($0.01 < p < 0.05$) and pairs of cells with bolded c:s those with $p < 0.01$. Index (timing) variables were square-root transformed before analysis except index at maximum withers that was analyzed squared.

cannon ROM compared to outer limbs: the forelimbs when ridden (inner vs outer 62° vs 63°, $p = 0.009$) and the hind limbs both when unridden (inner vs outer 53° vs 56°, $p < 0.0001$) and ridden (inner vs outer 52° vs 57°, $p < 0.0001$).

Withers and S3, and cannon angle timing variables are summarized in Table 6, presented on a stride percentage scale to facilitate comparisons of results between Table 4 (step percentages) and Table 5 (stride percentages). For midline variables, the same number of "*" within a row indicates significant differences. For example, in unridden walk the timing of withers minima (but not maxima) differed between inner and outer steps,

**Table 5 Cannon sagittal pendulation models.**

| Variable | Limb | Ridden[a] | Estimate | SE | (a) same limb unridden/ridden | | | | (b) inner/outer in same condition | | | | (c) comparison to straight | | | |
|---|---|---|---|---|---|---|---|---|---|---|---|---|---|---|---|---|
| Cannon sagittal range of motion (°) | Inner fore | 0 | 62.59 | 1.10 | – | | | | – | | | | c | | | |
| | Inner fore | 1 | 62.08 | 1.10 | – | | | | | c | | | | | | |
| | Inner hind | 0 | 52.68 | 1.10 | | – | | | | | c | | | c | | |
| | Inner hind | 1 | 52.26 | 1.10 | | – | | | | | | c | | | | |
| | Outer fore | 0 | 62.24 | 1.10 | | | – | | – | | | | | | c | |
| | Outer fore | 1 | 62.68 | 1.10 | | | – | | | c | | | | | | |
| | Outer hind | 0 | 56.31 | 1.10 | | | | – | | | c | | | | | c |
| | Outer hind | 1 | 56.53 | 1.10 | | | | – | | | | c | | | | |
| | Straight fore | 0 | 64.80 | 1.09 | | | | | | | | | c | | c | |
| | Straight hind | 0 | 54.57 | 1.09 | | | | | | | | | | c | | c |
| Timing of maximum protraction (percent of stride) | Inner fore | 0 | 27.64 | 0.37 | – | | | | – | | | | – | | | |
| | Inner fore | 1 | 28.25 | 0.37 | – | | | | | c | | | | | | |
| | Inner hind | 0 | −0.34 | 0.37 | | – | | | | | c | | | – | | |
| | Inner hind | 1 | −0.34 | 0.37 | | – | | | | | | c | | | | |
| | Outer fore | 0 | 28.43 | 0.37 | | | – | | – | | | | | | – | |
| | Outer fore | 1 | 29.10 | 0.37 | | | – | | | c | | | | | | |
| | Outer hind | 0 | −2.66 | 0.37 | | | | – | | | c | | | | | c |
| | Outer hind | 1 | −2.52 | 0.37 | | | | – | | | | c | | | | |
| | Straight fore | 0 | 28.06 | 0.37 | | | | | | | | | – | | – | |
| | Straight hind | 0 | −0.92 | 0.37 | | | | | | | | | | – | | c |
| Timing of maximum retraction (percent of stride) | Inner fore | 0 | 2.19 | 0.51 | – | | | | c | | | | c | | | |
| | Inner fore | 1 | 3.01 | 0.51 | – | | | | | c | | | | | | |
| | Inner hind | 0 | 14.69 | 0.51 | | – | | | | | c | | | – | | |
| | Inner hind | 1 | 15.33 | 0.51 | | – | | | | | | c | | | | |
| | Outer fore | 0 | 3.10 | 0.51 | | | – | | c | | | | | | – | |
| | Outer fore | 1 | 3.74 | 0.51 | | | – | | | c | | | | | | |
| | Outer hind | 0 | 12.83 | 0.51 | | | | – | | | c | | | | | c |
| | Outer hind | 1 | 14.39 | 0.51 | | | | – | | | | c | | | | |
| | Straight fore | 0 | 4.01 | 0.51 | | | | | | | | | c | | – | |
| | Straight hind | 0 | 15.38 | 0.51 | | | | | | | | | | – | | c |
| Maximum protraction angle (°) | Inner fore | 0 | 20.93 | 1.08 | – | | | | – | | | | – | | | |
| | Inner fore | 1 | 20.35 | 1.08 | – | | | | | – | | | | | | |
| | Inner hind | 0 | 37.42 | 1.08 | | – | | | | | – | | | – | | |
| | Inner hind | 1 | 36.92 | 1.08 | | – | | | | | | c | | | | |
| | Outer fore | 0 | 21.26 | 1.08 | | | – | | – | | | | | | – | |
| | Outer fore | 1 | 20.26 | 1.08 | | | – | | | – | | | | | | |
| | Outer hind | 0 | 38.53 | 1.08 | | | | – | | | – | | | | | c |
| | Outer hind | 1 | 38.94 | 1.08 | | | | – | | | | c | | | | |
| | Straight fore | 0 | 21.37 | 1.02 | | | | | | | | | – | | – | |
| | Straight hind | 0 | 36.51 | 1.02 | | | | | | | | | | – | | c |

| Variable | Limb | Ridden[a] | Estimate | SE | (a) same limb unridden/ridden | (b) inner/outer in same condition | (c) comparison to straight |
|---|---|---|---|---|---|---|---|
| Maximum | Inner fore | 0 | −41.37 | 1.09 | – | – | c |
| retraction | Inner fore | 1 | −41.74 | 1.10 | – | – | |
| angle | Inner hind | 0 | −14.97 | 1.09 | – | c | c |
| (°) | Inner hind | 1 | −15.34 | 1.10 | – | c | |
| | Outer fore | 0 | −40.92 | 1.09 | – | – | c |
| | Outer fore | 1 | −42.03 | 1.10 | – | – | |
| | Outer hind | 0 | −17.71 | 1.09 | – | c | – |
| | Outer hind | 1 | −17.19 | 1.10 | – | c | |
| | Straight fore | 0 | −43.27 | 1.05 | | | c   c |
| | Straight hind | 0 | −17.90 | 1.05 | | | c   – |

**Note:**

Least square means estimates and standard errors (SEs) for cannon sagittal pendulation variables ($n = 4,232$, $n = 4,015$, $n = 4,210$, $n = 4,232$ and $n = 4,232$, respectively). Models include data from 10 horses, unridden and ridden by 5 riders, in total 14 trials. Pairs of cells with '–' in the same column indicate non-significant comparisons ($p \geq 0.05$) and pairs of cells with bolded c:s those with $p < 0.01$.
[a] Unridden (0), ridden (1).

**Table 6 Timing of minima/retraction (normal font weight) and maxima/protraction (bolded) values on the circle in unridden and ridden walk.**

| | | | | | Percentage of stride | | | | | |
|---|---|---|---|---|---|---|---|---|---|---|
| **Unridden walk** | 0 < 10 | 10 < 20 | 20 < 30 | 30 < 40 | 40 < 50 | 50 < 60 | 60 < 70 | 70 < 80 | 80 < 90 | 90 < 100 |
| Withers | | **12.5** | | 39.3* | | | **62.6** | | 87.6* | |
| S3 | | 14.9** | | | 42.6*** | | 64.1** | | | 91.3*** |
| Inner fore | 2.2# | | **27.6** | | | | | | | |
| Outer fore | | | | | | 53.1# | | 78.4 | | |
| Inner hind | | | | | | | 64.7## | | | 99.7### |
| Outer hind | | 12.8## | | | 47.3### | | | | | |
| **Ridden walk** | | | | | | | | | | |
| Withers | | **12.3*** | | 38.7** | | | **62.9*** | | 86.5** | |
| S3 | | 16.1*** | | | 44.1**** | | 65.2*** | | | 94.1**** |
| Inner fore | 3.0# | | **28.2##** | | | | | | | |
| Outer fore | | | | | | 53.7# | | 79.1## | | |
| Inner hind | | | | | | | 65.3### | | | 99.7#### |
| Outer hind | | 14.4### | | | 47.5#### | | | | | |

**Note:**

All timings are presented as percentages of a stride with maximum inner hind limb protraction defining start and end of the stride. For between-step comparison for the withers and S3 the same numbers of '*' within a row denote significant comparisons, while the comparison without stars is non-significant. For limb variables the same number of '#' across rows demonstrates significant differences for inner/outer limb comparisons. Results recalculated from Tables 4 and 5, dividing midline variables timings by two (converted from step to stride percentages) and adding or subtracting 50% when appropriate.

occurring relatively later for the inner step (39.3% * 2 > 87.6%, first row of Table 6). For limb timing variables, the same number of "#" across rows shows statistical differences. For example, when ridden maximum protraction occurred earlier for the inner forelimb compared to the outer forelimb ($p = <0.0001$, 28.2% + 50 < 79.1%, figures from the third and fourth last rows of Table 6).

### Stride duration

In the stride duration model, circle/straight within unridden/ridden and hind limb cannon protraction-retraction ROM were both associated with $p < 0.0001$. Stride duration was shorter for straight compared to on the circle, but there was a minuscule ($p = 0.30$) difference between when ridden and unridden. The coefficient for hind limb cannon ROM was −0.011 s/°, suggesting that stride duration decreased with increasing cannon ROM.

## DISCUSSION

### Straight vs circle

In the current study the intent was to investigate the effects of circular movement at walk with minimal influence from other factors, such as the handler or a rider. It could be assumed that values for straight line would be in-between inner steps and outer steps on the circle, but for some variables estimates for straight-line walk fell outside that range. S3 ROM, and forelimb cannon ROM and forelimb maximum retraction were all greatest for straight-line walk, the latter possibly indicating increased carpal flexion. Straight walk was measured on concrete, using an aisle to enhance straightness of the horse, while walk on the circle was measured on soft surface. Because of this, a surface effect may (also) be involved. Speed, stride duration and stride frequency may affect cannon ROM (*Clayton, 1995*). Stride duration was slightly longer on the circle compared to straight, while similar between unridden and ridden conditions (Table 2). All models were controlled for stride duration (selected results are reported in Table 2). For variables where straight-line values were within the inner–outer range, it varied if these differed significantly from both inner and outer limbs/steps, or from only one of these. Hind limb cannon ROM is an example of the former: compared to straight the outer hind had larger ROM, while the inner hind had smaller ROM. Larger outer hind ROM was found together with larger protraction, and smaller ROM for the inner hind limb found together with less retraction, compared to straight. Further, with regard to changes in timing of maximum protraction and maximum retraction, the inner forelimb (earlier retraction) and the outer hind limb (earlier protraction and retraction) differed from straight. This suggests that adaptation to circular movement is not evenly distributed between contralateral limbs. The vertical excursion of the withers and S3 seem to have been more generally affected by the circle, since for all significant comparisons, minima/maxima occurred later on the circle, regardless whether inner or outer step.

### Inner vs outer

In the current study, there were several significant differences between inner and outer limbs, and inner and outer steps for midline variables. Without rider, a majority of the significant differences were related to the hindquarters. With rider, there were also several significant differences in the forehand variables. The outer hind limb had larger cannon ROM and maximum retraction angle, compared to the inner for both conditions.
Outer hind maximum retraction occurs during the first part of the inner hind limb step. A larger retraction angle could indicate a larger longitudinal spread between the two hind limbs at this instant, explaining the lower S3 minimum, and larger S3 vertical ROM

during the inner step, compared to the outer step. When ridden, the outer forelimb similarly had relatively larger cannon ROM. Along with this, withers vertical ROM was larger from outer forelimb midstance to inner forelimb midstance, compared to the other stride cycle half, and withers minimum position was deeper during early stance of the inner forelimb. These findings point towards a close coupling between limb spread and the vertical movement and minimum position of the croup (S3), and the same is suggested for the withers. Such coupling is predicted by the inverted pendulum model for the biomechanics of the walk (*Kuo, 2007*).

Hind limb maximum protraction and maximum retraction both occurred earlier for the outer compared to the inner hind, and when ridden the outer forelimb reached maximum protraction relatively later. Timing differences between inner and outer limbs resulted in an asymmetry in durations between maximum protraction of sequential limbs, somewhat more pronounced with a rider compared to without. Using the results from Table 6, durations in percentage of the stride when horses were unridden (ridden) were:

27.9% (28.5%) from inner hind limb to inner forelimb;
19.7% (19.3%) inner forelimb > outer hind limb;
31.1% (31.6%) outer hind limb > outer forelimb;
21.3% (20.6%) outer forelimb > inner hind limb.

Forelimb cannon maximum protraction and retraction occur approximately 2% before toe-on and 12% after toe-off, and hind limb cannon endpoints occur approximately 8% before toe-on and 8% before toe-off, based on *Hodson, Clayton & Lanovaz (2000, 2001)*. Because these offset percentages are not the same for forelimbs and hind limbs, timing for protraction and retraction of the cannons will misrepresent the hoof-beat rhythm. The rhythm is expected to be relatively even four-beat in (equine) walk. Figure 2 shows cannon protraction-retraction angles for one horse. It can be seen that forelimb maximum protraction is soon followed by maximum protraction of the diagonal hind limb (<25% of the stride between peaks), and comparing fore- and hind limb retraction are even closer. To approximate durations between hoof ground contacts (toe-on), the hind limb-to-forelimb intervals above should be adjusted by subtracting 6%, and forelimb-to-hind limb intervals by adding 6%. Following such adjustment, the estimates above would suggest that the horses tended to walk with lateral couplets, which is in accordance with previous studies (*Clayton, 1995*; *Weishaupt et al., 2006*). These studies further concluded that the rider can influence the evenness of the walk rhythm.

The timing of withers and S3 vertical excursions were also affected by the circle. S3 minimum position, which occurs during early hind limb stance, was relatively delayed for the inner step compared to the outer step, both with and without rider. This may relate to that the inner hind limb reached maximum protraction later, which would suggest that hoof landing was relatively delayed for the inner hind limb. Withers minimum was similarly delayed for the inner step vs the outer. The kinematics behind this is more difficult to deduct. Inner forelimb maximum protraction occurred relatively earlier, not later. It could not be approximated when the outer (or inner) forelimb was at its most

retracted position, since the forelimb cannon will continue to retract as the carpal joint flexes. Maximum forelimb cannon retraction occurs at about 70% of the stride, which is in-between toe-off of the forelimb, at 64%, and maximum flexion of the carpal joint, at 76% of the stride (*Hodson, Clayton & Lanovaz, 2000*).

## Unridden vs ridden

In Warmblood horses walking on a treadmill, a relatively lower withers minimum during early stance of one forelimb, compared to the other, was associated with left-right differences in stance duration and limb loading both with and without rider. However, while asymmetries in hind limb stance duration and hind limb second vertical force peak were associated with withers asymmetry without rider (*Byström et al., 2018*), when horses were ridden withers asymmetry was instead associated with asymmetries in forelimb stance duration and forelimb first vertical force peak (*Egenvall et al., 2020*). In the current study, withers ROM was smaller for both inner and outer steps when ridden on the circle compared to without rider. Additionally, the withers minimum during early stance of the inside forelimb became significantly lower compared to the minimum for the outer limb/step. This suggests that adding a rider limits the vertical excursion of the forehand, and relatively more during the period from midstance of the inner fore to midstance of the outer fore, while the vertical movement of the hindquarters is less affected. The inner vs outer step difference in withers minimum, that appeared with a rider, might relate to that riders are often instructed to place somewhat more weight on the inside vs outside seat bone when riding on the circle (*Hess et al., 2012*). Rider weight has been shown to influence ground reaction forces of the forelimbs more than on those of the hind limbs in walk and trot (*Schamhardt, Merkens & Van Osch, 1991*; *Clayton et al., 1999*). Forelimb maximum retraction has been shown to increase when the horse carries weight in both walk and trot, measured on straight line (*DeCocq, Van Weeren & Back, 2004*). As mentioned previously, this variable could not be approximated in the current study, since the forelimb cannon maximum retraction angle occurs when the limb is flexing during swing. However, in our horses, walking on the circle, there were no significant differences in any of the cannon protraction–retraction variables between unridden and ridden conditions (comparing each limb, fore- or hind, inner or outer).

In addition to decreased withers ROM, our IMU-data suggest that riding the horse on the circle disrupted the biphasic vertical motion pattern of the forehand (the withers) (Figs. 2 and 3). The thorax is suspended in the thoracic sling and supported by the serratus muscle. The scapula has been shown in one horse to move relative to the trunk in all three planes (dorso-ventral, medio-lateral and cranio-caudal) during walking (*Lawson & Marlin, 2010*). It has not been studied whether these movements are influenced by the addition of a rider, but it seems reasonable that the increased weight load will have some effect. However, personal observations from preliminary high-speed video data suggest that the effect of the rider on the vertical motion of the withers is less than the IMU-data are suggesting.

### Benefits and limitations

There was only 10 horses and 5 riders. The horses were relatively small, with a maximal height of 158 cm, and included one pony. Data were evaluated from an experiment designed to mimic normal training sessions (except for adhering strictly to the 10 m circle). Since the riders rode almost without rein contact or using leg aids, the rider's influence was essentially limited to the actual weight load, and the riders were of similar weight; the results are therefore less likely to be influenced by individual variation between riders. The equipment varied (Table 1), including that one horse was ridden bareback because this horse had only been ridden bareback for several years; the same horse was also ridden bitless, for the same reason. Despite considerable variation in horse characteristics, there were still many significant effects at group level, which suggests good general applicability of our findings. Stride duration was somewhat shorter at straight-line walk and we decided to control for this in the analysis, since cannon angles change in association with stride duration (Clayton, 1995). We did not record speed because this would have relied on that all horses followed the 10 m circle exactly. Thus, the analyses were not controlled for speed, but because of the coupling between speed and stride duration (Weishaupt et al., 2010) this was achieved indirectly. In the extraction of strides, we aimed to select sequences of regular motion, hence fewer strides were selected from some exercises for some horses. For example, trials from horses with more unsteady head movements yielded fewer strides. Additionally, for midline variables, especially for the withers, the number of strides used in the analysis was lower than the overall selection (as reported in Table 3). Given the chaotic appearance of the withers signal when horses were ridden (Fig. 3), comparing withers variables ridden vs unridden is likely the weakest part of our analysis. We recommend that future studies on withers vertical excursion in ridden horses use high-speed video. Comparing our results for cannon protraction–retraction to previously published results from high-speed video data for unridden straight walk, ROMs are similar but maximum protraction and retraction angle values are not (Hodson, Clayton & Lanovaz, 2000, 2001). The discrepancies may be due to failure to align the IMUs/optical markers with the long axis of the cannon bone (possible misalignment led us to compare differences for the same limb between conditions, for example, the left hind as inner vs outer limb, by adding an extra random effect to this model). We finally note that the purpose of this study of this study was not to investigate left-right differences or side preferences on an individual basis.

## CONCLUSIONS

The outer forelimb had greater cannon ROM in ridden walk, as had the outer hind limb both with and without rider, compared to the inner limb. Larger croup vertical ROM during the inner step appears to be coupled to increased retraction of the outer hind limb. For walk on the circle, the presence of a rider mainly influenced the movements of the forehand: withers vertical ROM decreased and additional significant inner–outer differences appeared for withers and forelimb variables. Knowledge of the magnitudes and timing of the horse's movements on the circle in unridden and ridden walk may stimulate

riders to educate their eye and feel for the (correct) execution of circles. Such knowledge may also encourage further studies on walk kinematics, for example, how primary effects of circular movement interact with other factors such as active influence from the rider, natural horse asymmetries/sidedness, or lameness.

## ACKNOWLEDGEMENTS

We thank the riders for participating in the study.

### Funding

Funding for the experiment was provided by a Career Grant provided to Agneta Egenvall from the Swedish University of Agricultural Sciences. The funders had no role in study design, data collection and analysis, decision to publish, or preparation of the manuscript.

### Grant Disclosures

The following grant information was disclosed by the authors:
Swedish University of Agricultural Sciences.

### Competing Interests

The authors declare that they have no competing interests.

Hanna Engström is co-founder of Ekeskogs Riding Academy.

### Author Contributions

- Agneta Egenvall conceived and designed the experiments, performed the experiments, analyzed the data, prepared figures and/or tables, authored or reviewed drafts of the paper, and approved the final draft.
- Hanna Engström conceived and designed the experiments, performed the experiments, authored or reviewed drafts of the paper, supervised correct execution of the exercises, and approved the final draft.
- Anna Byström conceived and designed the experiments, analyzed the data, authored or reviewed drafts of the paper, and approved the final draft.

### Human Ethics

The following information was supplied relating to ethical approvals (i.e., approving body and any reference numbers):

According to Swedish law this kind of experiments, with minimally invasive measurements during activities normally undertaken by the subjects, does not require human ethical permit, given that GDPR standards are met.

### Data Availability

Raw data, code and full statistical (SAS) output are available in the Supplemental Table.

## Supplemental Information

Supplemental information for this article can be found online at http://dx.doi.org/10.7717/peerj.10354#supplemental-information.

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
