# Peer review of "Kinematic effects of the circle with and without rider in walking horses"

_PeerJ, doi:10.7717/peerj.10354_

## Round 0.1 · original submission · Major Revisions

Thank you very much for your interesting manuscripts. This manuscript has a high potential to publish in PeerJ. However, concluded from two reviewers, this manuscript requested a revision.

·

Basic reporting

The manuscript requires further work to improve the clarity and written English.

Experimental design

This study is original research and within Scope of the journal. The research question is not well defined, although relevant. It is not well stated how the research fills an identified knowledge gap. Methods require further detail.

Validity of the findings

Findings are valid. All underlying data have been provided. Please see general comments in relation to more detailed feedback.

Additional comments

The background section of the abstract is too short. There needs to be a better explanation of the reason for performing this study.
Line 20: curved, not bent
Line 27: laterally on cannons is vague. Fore and hindlimbs? Is this in brushing boots? If so, state.
Line 28: Was this split for a full stride using the left or the right hindlimb?
Line 29: extracted?
Line 30-31: It is not clear what step and cannon (limb) variables actually are.
The results section is too long and does not highlight well the key results from this study. There is a need to tie these into the reasons for performing this study better.
The conclusion is not very strong. Once key results are highlighted better the conclusion needs to provide a more justified reason for studying circles at walk. It that the only conclusion coming from this work? Why? What findings indicate that more research on circles will be useful?
Line 57-58: This results in that the….poor grammar. This results in out of phase movement of the fore- and hindquarters.
Line 60-61: ‘Even if the walk is slow, the movement pattern is complex.’ I am not sure what this statement is saying. Is a fast walk pattern not similarly complex then?
Liner 61-63: Would it not be better to just describe the motion of the saddle in walk, instead of comparing to trot.
Line 64: The rider does not maintain a four-beat rhythm, the horse does. Please reword.
The first paragraph needs additional references. Also, I assume that all of these descriptions are in a straight line?
Paragraph 2 contains no references. Also, what is important in relation to studying the horse-rider interaction during walking. Yes, the horse may be asked to bend to the left or right by the rider, but what is ‘important’ about this?
Line 75-83: This is a better paragraph showing what has already been found during turning at walk in horses. But is leads me to wonder…so what? What does this mean? What is important about it? Why study the walk during turning further if we already know this?
The aim has no mention of the difference between ridden and in hand turning. Why only hindlimb steps? The second statement is just about measurements. You need to justify why we need to know this and also you need to include hypotheses. The aim is also quite vague.
Line 93: for not since
Line 95: Can you be more specific. Lameness scale scores?
What type of horses/riders? What level? Dressage or not?
Line 104: Age range of riders, not specific ages.
Line 105: For not since
Lines 112-113: Please provide more specific anatomical detail in relation to sensor placement. If the cannon sensors were in brushing boots this should be stated, together with whether they were attached to all four limbs. Cannon’s is also rather layman language. Please improve.
Line 113: Sign convention should be clearly stated in a new sentence.
Line 116: I don’t understand what this relates to. (other IMU-recorded data not used in the study)
Line 118-121: Where was the camera placed when recording in a straight line? Also if the sensors were tapped before putting them on the horse I assume you continued recording throughout the time that they were placed on the horse? Why on a stool? Why not placed on the horse and then tapped?
Line 127: Are AE and HE researchers?
How many days did each horse perform the tests? I assume this is because some riders rode more than 1 horse? Perhaps a table describing this in more detail would be useful.
I am finding the description of ridden/lunged/left/right confusing. From what you have written I assume they were all led and lunged first, then ridden afterwards. If they were lunged first to the left, then they were ridden first to the right? Is this what you mean? Why?
Why did you decide to use minimal contact between the horse and rider? I expected that you might have investigated the difference in ridden and unridden conditions where the rider rode normally. By not doing this, I expect you are just looking at the effect of the rider sitting on the horse (so essentially not much more than a mass increase), rather than the horse-rider interaction? This should come through in the aims and hypotheses much more clearly.
Line 148: Did you assess the straight line data for L/R symmetry? I can see the logic in splitting using the inside hind on a circle, but if the L/R data was not symmetrical (and you didn’t assess the horses quantitatively for soundness), then you may be introducing errors due to asymmetry when splitting the circle data.
Line 152-163: It is not clear or justified as to why you have chosen these variables, why they are split in the way you have split them and why you have constrained certain data. You may need to include some graphical evidence or at least a better justification for doing this. S1 appears for the first time here. I think you used croup previously?
I have to say that reading the statistical analysis gives me the impression that you have thrown everything into the model with the justification that this is exploratory, even though some studies have already investigated walk kinematics. The statistics should map to the aims and hypotheses of the study more clearly and the reasons for your modelling should be better justified.
Line 213: What is a trial?
Line 214: were not was
Figures and tables should be in the order they first appear in text.
Line 228: 39.3%*2>87.6%, figures first row of Table 5 what does this mean?
When stating significance in text please include p values as a minimum.
Please be consistent when reporting vertical excursion. Either mm or m, but not both.
Line 262-270: It is not clear whether all of these results are significant.
There is quite a large amount of text, tables and diagrams illustrating the results. I would suggest you include just the highlights in the text and allow the tables and figures to report the majority of these findings. There should not be excessive duplication.
The discussion should begin with a summary of the study, highlighting any main findings and stating whether the hypotheses may be accepted or rejected. Please include.
The discussion should not be merely a comparison of your data to other studies. It should focus on what is important about your findings. You need to critique other studies in relation to the difference in walk kinematics in a straight line and on a circle, what are the differences? How does this relate to other studies? Why are they different? Different equipment, methods, horses etc. It is not about presenting values from other studies and just comparing them to yours. It is more about why you have found what you have found. Also, do not split by measurement variables. Split by conditions (i.e. ridden/unridden, straight/circle) and compare the ‘whole horse’, for example when you have a change in max protraction in the forelimb, how does this influence the vertical displacement of the withers and why would a rider being on top influence that? The later sections are improved, but mainly focus on what the differences are between your data and other studies and not why these differences are likely to have occurred and what that means in the wider context of the equestrian world (i.e. for riders, for lameness evaluations etc).
Table 2: The legend for this table does not match all of the data? You have included ROM estimates? This is not detailed in the legend. Also, your min and max estimates and ROM for pro-retraction seem to be much smaller than pro-retraction values recorded in walk?
Table 3&4: What does 1 and 0 mean? What comparisons are the colours indicating? Define non-significant. Why are some of the squares blank?
Figure 2: The arrows on the diagram are not always in the right place and appear to be sketched. Please improve the quality of this figure and label the axes for angles on one side and vertical displacements on the other. Include units.
Table 5: The colour scheme seems to indicate the same colours between comparisons for all data? Do you need to state here that values with no colour are not significant. i.e. there is no difference between the unridden and ridden condition when the cells are not coloured.
Figure 3: Please graph the ridden, unridden and straight conditions on the same graph for that limb. S1 and withers data can be graphed separately. Also, this should be inner and outer, not left and right.
Figure 4: As for Figure 3. In addition, include inner and outer fore. Inner and outer hind.

Reviewer 2 ·

Basic reporting

The manuscript requires a revision to improve manuscript.

Experimental design

The research question and hypothesis are not well defined, although relevant. Methods and Discussion are require further detail.

Validity of the findings

Finding are valid.

Additional comments

thank you for the interesting information. However, it need to re-write to academic writing.
According to this study show many data, I would recommend make it compact and pick the data that
answer your hypothesis. Please select the importance information and present in graph or table
which should clear and simple. I commend the author which should be improved before acceptance.
1. Line 4 Agneta Egenvall 1
,* what is refer to, if it mean Corresponding Authors : you should
to change to Agneta Egenvall 1
*, and put the * infront of Corresponding Authors
2. Abbreviation Line 28 at the Abstract: ROM, It would be good for reader to understand please
give full name if it at the first page or first time in the paragraph. Example; range of motion
(ROM)
3. Abbreviation Line 43, 44 what is the S1?
4. Abbreviation Line 80 what is the EMG refer to?
5. Abbreviation Line 95 a veterinarian (AE), Is it necessary?
6. Abbreviation Line 127,139 AE, HE, AB Is it necessary? Do you what show a veterinarian(AE)
is the author of the study and also the lameness examiner, rider and video recorder ? How is
importance?
7. Abbreviation Line166 what is Proc MIXED?
8. Line170 What do you mean the rider*horse?
9. Abbreviation Line171 What is pdiff?
10. Line 183 What do you refer to: horse*rider*exercise?
11. Line 195 what do you refer to: horse*rider*horse side?
12. Please check the reference style in the Introduction and Discussion. There are variety of
reference example at Line 63 (Byström et al., 2010), Line75 Chateau, Degueurce &
Denoix (2005) Line79 Wakeling et al. (2006)
13. Please use words in consistency
Example: Line 85 Forehand / Wither / forelimb? , Line 86: Hind quarters/ hind limb? Line88
Croup/ S1 / Sacrum ?
14. Please make Objective of this study is clearer. How many objective of this study?
15. For Materials and methods : Line95 This study use subjective lameness examination by a
veterinarian. Do you think only a veterinarian can be bias? It has been reported in previews
study from Keegan. Why don’t you use Objective tool for lameness examination because in
this study you used inertial measurement units (IMU) to measure? Do you think lameness
horse might give you difference results?
16. For Materials and methods : Line 108 ( Table1)
It would be useful for the reader if you show the detail weight of saddle plus weight of rider
rather than Duration with the owners (I guess relation of that horse and the owner).
17. For Materials and methods: Line 108 (Table1) Breed of horse in this study. Do you think
difference breed or difference activity is might have difference of character to walk? Type of
small horse: Do you think the size of horse might affect with the results?
18. For Materials and methods: Equipment Line 112 Inertial measurement units (IMU)
(https://x-io.co.uk/ngimu/), Line 118 video-recorded (Sony FDR-AX53), Line 140
(Matlab version 2019b, The MathWorks® Inc., Natick, MA, USA) , The details of
these equipments should give more details for example IMU: What is the company,
program version have been used. The details should declare conflict of interest, support. The
details should be at additional part after reference section. Please check with the Journal
format. For each Tool will give difference value. It would be better if this study classify the
main parameters that obtained from which tool. ROM in degree, Vertical translation of the
wither and croup in mm. and give interpretation why is minus in degree.
19. For Materials and methods: Study design Line123 Do you think speed of walk might be effect
to ROM in the circle and straight-line? How do you control speed?
20. For Materials and methods: Study design Line123 Do you think natural walk and active walk
during longing and ridding might be effect with ROM?
21. Statistical analysis: Do you use only descriptive analysis and present in percentage?
22. Results Line 213, Refer to the study design each horse had 6 conditions: straight –line with
and without ridden, Left circle with and without ridden, Right circle with and without ridden.
In Total (10 horses) should be 60 Trials. Why in the results show 14 trails? How comes of
this number?
23. Results Line 211-331, after report the descriptive results of the horses and riders should
report in order what have done in the study design. Especially the data that fit with the
objective. For example, the data focused on vertical translation of the withers and the
croup, and ROM, minima, maxima and their timing for sagittal cannon angles in walk,
approximating limb protraction and retraction.
24. Results Line 212-221 why the range of stride of each condition was variety, especially
minimal range of stride? Do you think is that a good represent of your study?
25. Results Line 223 Academic writing need content in text of each new paragraph then refer to
Table or figure. This need to re-write “Tables 3 and 4 show model results that are further
explained below”.
26. Results Line 211-331 and Discussion Line 333-501 Need grammatical prove, Repetition, too
much additional information in the sentence.
Example: Line 337,339 Our results, Our
Line:354 was 36% for straight line (0.92+(-15.38)+50 [because of step adjustment]).
Line: 428-429 the outer hind limb differed from straight (earlier protraction and retraction),
while it was the inner forelimb that changed (earlier protraction).
Line: 440 increased first vertical force peak for the (same) forelimb in early stance
Line: 449 In the current study there was, however, no significant differences
Line 498-499 We recommend use of high-speed video for detailed studies of withers vertical
excursion in ridden horses. We also note that
27. Table and Figure
Table2. Title of this table is too long and it need to separate data into 2 tables
Table3. Title of this table is too long and too much explanation. What is the meaning of BTb
Colum? What is CompC
refer to ?
Table4. The subtitle in Variable Colum are duplicate. Please check Timing of maximum
retraction (percent of stride), It should be Maximum and Minimum
Figure 1 . Show a green sensor on the right forelimb which is not correlate with the text Line
113 that laterally on the cannons of all four limbs. It can make reader confuse.
Figure 2. I would expect to see how is the trend vertical movement of wither of each trial
compare with and without ridden
Figure 3. Is it croup or Sacrum or S1 vertical position? From the graph picture I got the
information that the horse without ridden movement sacrum more freely than ridden
condition is it the thing that you would like to represent?
Figure 4. Is this graph compare trend of outer and inner limb in circle ? Why straight line still
have outer and inner, How comes?
Line 235 what is the Figures S1-S3 that refer to? Where is it?
28. Conclusion: What is the take home massage for reader? From your study what is the impact
for sport rider, veterinarian and owner?

Annotated reviews are not available for download in order to protect the identity of reviewers who chose to remain anonymous.

---

## Round 0.2 · Minor Revisions

We need a minor revision prior to acceptance. Please revise your manuscript according to the review.

·

Basic reporting

Much improved. Just a few small changes suggested.

Experimental design

No comment

Validity of the findings

No comment

Additional comments

Abstract Conclusion. Overall the outer hind limb had greater ROM in pro-retraction,
‘and with a rider also the forelimb.’ Is this ….and there was an increase in pro-retraction ROM for the outer forelimb with a rider.
Line 73: these events coincide
Line 82: gait is missing
Line 128: angular ROM of the cannon bones
Line 546 and others: toe-on? Do you mean landing or foot strike?
Line 597: might be because riders
Line 612: disrupted the biphasic vertical motion pattern of the forehand (the withers)
Line 639: evaluated not evacuated
Line 652: extraction not evacuation

---

## Round 0.3 · accepted · Accept

Thank you for the revision of the manuscript, now your manuscript is ready to publish.